# Fostering Digitalization of Construction Projects through Integration: A Conceptual Project Governance Model

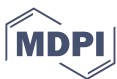

**Zhixue Liu [1], Ronggui Ding [1], Zheng Gong [2] and Obuks Ejohwomu [2,*]**

1 School of Management, Shandong University, Jinan 250100, China; zhixue@mail.sdu.edu.cn (Z.L.); ding_rgui@sdu.edu.cn (R.D.)
2 Department of Mechanical, Aerospace and Civil Engineering, The University of Manchester, Manchester M13 9PL, UK; zheng.gong-7@postgrad.manchester.ac.uk
* Correspondence: obuks.ejohwomu@manchester.ac.uk

**Abstract:** The construction industry has fared poorly in the process of digital transformation, while the main challenge is the digitalization of construction projects. Changes in project management approaches are urgently required in construction organizations to better align digital technology and organizational conditions. However, little literature has explored the pivotal role of the project management approach from an organizational perspective. To fill this gap, this research investigates ways of using a project governance model for integration to promote the digitalization of construction projects through a case study. The three integration dimensions, namely stakeholder integration, lifecycle integration, and project management knowledge integration, are identified, and governance elements under each dimension are displayed—and further stratified—based on the three levels of the governance model, including institutional level, organizational level, and behavioral level. The logical relationship between elements and their roles in project digitization is finally summarized. The developed conceptual model will provide an approach for construction enterprises to promote project digitalization.

**Keywords:** digitalization; construction projects; integration; project governance





## 1. Introduction

The industry 4.0 era is undergoing disruptive changes caused by new technologies, while data and technology are significantly improving people's work efficiency [1]. Within this context, adopting and applying digital technologies has become an inevitable choice for various industries and enterprises to sustain competitiveness [2–4]. However, the construction industry has fared poorly in the process of digital transformation, ranking at the bottom of the 22 industries in terms of digitization level [5]. As the construction industry is project-based, the digital transformation of the construction industry or construction enterprises is considerably determined by successfully digitalizing construction projects. However, the digitalization of construction projects has been deemed a challenging task, due to the unique and temporary nature of construction projects, as well as their increasing scale and complexity [6]. Therefore, realizing the digitalization of construction projects is of great significance to foster the digital transformation of the construction industry.

Digitalization in the construction industry commonly refers to using digital technology to fundamentally change construction processes, thereby improving construction output and productivity to achieve enhanced project outcomes and better client satisfaction [7]. Currently, various digital technologies—BIM, autonomous robots, cloud computing, 3D printing, the Internet of Things (IoT), augmented reality (AR), and big data analytics—have been introduced, and they have brought progress to the digitalization of construction projects [8,9]. They are expected to bring transformation in project delivery and significantly improve efficiency and productivity, yet they are still far from reaching their full potential [10,11]. For example, BIM, one of the most widely adopted digital technologies in

construction projects, has great potential to add economic, social, and environmental value to projects by deepening collaboration between stakeholders and integrating information in the project lifecycle [12]. In practice, its adoption among participants and application in project tasks is limited [13,14].

In essence, digitalization is a socio–technical system whose effectiveness depends on the degree of coupling between the social and technical aspects of the system [15,16]. The simple adoption of digital technologies in organizations without the corresponding technology-mediated organizational shift results in the underperformance of digitalization [17,18]. To exert transformational digitalization on the whole organization, organizational issues need to be considered in the design and implementation of digital technology. Integration is considered a primary characteristic of digital technology, which can support integration between and within organizations [19]. In construction projects, digital technology should promote project integration and optimize project outcomes [20]. However, recent research has reported that current organizational factors, such as project stakeholders' skepticism and resistance to digitalization, lack of clear benefits, and stakeholders' lack of digital experience and knowledge, significantly restrict the integrated function of digital technology in construction projects [21]. These unfavorable organizational conditions limit the application of digital technologies and obstruct project integration. It is, therefore, necessary to reduce organizational barriers by transforming the decentralized project network into an integrated organizational structure and ways of working. Project governance addresses the organizational structures and processes, as well as project roles and responsibilities assigned to stakeholders and project structures [22]. It provides the approaches, authorities, accountabilities, and processes to define the objectives of projects, the means to achieve the objectives, and the control process [23]. Therefore, it can adjust the project organization network to realize integration in digitalization through governing relationships of various participants and forming inter-organizational coordination.

However, the existing research on the digitalization of construction projects has mainly been limited to the technical aspect, such as how to digitally express construction projects [24,25]. The social aspect of digitalization, especially the organizational shift of the project to facilitate the integrated functions of digital technology, is still poorly studied.

Therefore, the purpose of this study is to address the organizational barriers of digitalization and reduce the misalignment of project organizations and digital technology by adopting a project governance approach to improve the application performance of digital technology in construction projects. To achieve this, the following research questions are posed: (1) What is the relationship between project governance, project integration, and digitization? (2) How can project integration be achieved to provide a favorable environment for the application of digital technology through a project governance approach, thereby promoting the digitalization of construction projects? The remainder of this study is structured as follows: The literature on digitalization and integration in construction projects, project integration, and project governance is reviewed in the next section. After that, the research methodology is presented in Section 3. Section 4 reports the analysis process and the results of the case study. The discussion of the findings is presented in Section 5. Finally, the conclusions, limitations, and ideas for future research are presented in Section 6.

## 2. Literature Review

### 2.1. Digitalization of Construction Projects and Integration

#### 2.1.1. Digitalization in Construction Projects

Recently, the focus on the digital transformation of the construction sector has increased dramatically from both academia and practitioners, particularly with the impact of the COVID-19 pandemic and the rise of remote work. Commonly, the digital transformation process can be divided into three stages, namely digitization, digitalization, and digital transformation [3]. Digitalization is a process in which digital technologies are used to optimize business processes [11,26]. In the construction industry, digitalization is mainly

implemented on construction projects and is the requisite stage moving toward company and industry-wide digital transformation [27,28].

Scholars have interpreted digitalization in construction projects from different perspectives, such as innovation [29], change [30], or socio–technical systems theory [31,32]. The socio–technical systems theory has advantages in explaining the interdependence between the technical and social aspects of the digitization system [33]. Whether digitalization is viewed as an innovation or organizational change, a primary reason for its failure is an excessive focus on one aspect of the system, commonly technology, without analyzing and understanding the socio–technical interaction [34]. Therefore, the successful implementation of digitalization in construction projects requires both actual technological installation and social adaptations of the project organization network [15,35,36].

Researchers in the construction field have extensively studied the technical aspect of digitalization and proposed various digital solutions. These solutions are mainly based on BIM technology, combined with other digital technologies, to collect, analyze, and present data from different phases in the project lifecycle to support project management (PM). For instance, the integration of BIM with real-time data from IoT devices can apply to areas including construction operation and monitoring [37,38], health and safety management [39], and construction logistics and management [40]. Virtual reality (VR) and wearable technologies were considered to expand BIM to effectively manage workers' health risks and emergencies through pre-planning, education and training, and on-site monitoring [41,42]. Robotic systems and automation enabled by BIM were regarded as having great potential to improve construction productivity, reduce labor costs, and avoid injuries [43,44]. Unmanned aerial vehicles (UAVs) were also proposed to assess the project progress and perform compliance checks of geometric design models in conjunction with BIM [23]. Furthermore, the application of big data was introduced to support the management of projects and predict the performance of future projects through collecting, storing, and analyzing the massive volume of data in projects [42,45,46].

However, the implementation of BIM-based digitalization in construction projects encounters numerous challenges and setbacks, most of which lay on the organizational side rather than the technology itself. Sawhney et al. [47] pointed out that the conservative viewpoints of senior project leaders would lead to skepticism and resistance to change, resulting in a slow digitization process. Stakeholders' inconsistent attitude toward digitalization was also regarded as an obstacle to digitalization, which is caused by their differences in digital capabilities and willingness to digitalize [48]. Additionally, even though construction organizations can share their digital resources with digital partners to gain a better competitive advantage, improved project performance, and risk reduction, it is difficult to achieve in practice because of the poor definition of goals, trust issues, partnering risks, and investment cost [8,49]. However, little attention has been paid to organization-related features of digitalization, and there is a lack of research on addressing the organizational barriers to the digitalization of construction projects from the standpoint of PM.

### 2.1.2. BIM-Based Integration

BIM has been widely viewed as a revolutionary technology in the construction industry and plays a vital role in the digitalization of construction projects. It is a fundamentally different way of creating, using, and sharing building lifecycle data, and it can bring benefits to every aspect of the project lifecycle from planning to demolition [50–52]. Miettinen and Paavola [10] summarized four ambitions of BIM implementation: (1) all relevant data needed in the design and construction of a building will be included in a single BIM model or are easily available with BIM tools; (2) a tool for collaboration allowing new integrated ways of working through data interoperability; (3) being maintained and used throughout the lifecycle of the building; (4) considerably increasing the efficiency and productivity of the building industry. By combining with other digital technologies, BIM is further expected to support PM by facilitating integration in projects from three dimensions: stakeholder integration, PM knowledge integration, and lifecycle integration.

BIM has been shown by many studies to foster collaboration between stakeholders. By building a BIM-based digital platform, information can be shared between stakeholders in a unified and convenient way, both on-site and off-site [53,54]. This can promote communication and collaboration, thereby improving work efficiency; the knowledge and experience of participants can also be put into the project to contribute to the co-creation of value [55]. As for the integration of PM knowledge, some scholars indicated that BIM supports project integration management by integrating data from different PM knowledge domains [56,57]. By connecting functional subsystems with the BIM database, BIM can support the coordination of project schedule, cost, quality, resource, and other elements, simultaneously, to achieve optimal management of the whole project [58,59]. Using other digital technologies, such as IoT, the project data can also be collected and integrated in real-time to monitor and control project work [60]. Furthermore, BIM can also integrate data in the project lifecycle to support management and decision-making at all stages. This requires the continuous use and transfer of the BIM model between different actors to ensure that BIM functions throughout the project lifecycle. Based on this, BIM can integrate the management requirements at different stages of a construction project into the functional application of BIM and achieve efficient PM [61]. It can also support lifecycle decision-making by enabling data reuse in all stages [62].

## 2.2. Project Integration and Project Governance

### 2.2.1. Project Integration

The construction industry has long been deemed fragmented and unintegrated, which encourages adversarial relations, incurs conflicts between activities, and leads to productivity reduction and variability in project performance [63–65]. Therefore, project integration is believed to significantly improve project performance, which researchers have interpreted from different perspectives, such as coordinating processes [66], improving the integration of information and knowledge [67–69], promoting innovation [70], and managing risks comprehensively [71]. Project integration also plays an essential role in promoting the digitalization of construction projects, as it fosters inter-organizational cooperation and creates an environment for the exchange of digital resources between actors [65,72]. This cooperation across organizational boundaries reduces their learning costs in digitization and creates benefits for them by uniting the resource portfolios and activities of different actors, thereby increasing their acceptance of digitization [73]. Thus, a circular flow of information and resources between stakeholders can be formed to continuously identify and seize opportunities throughout the project, leveraging digital technologies to create more value for the client [74].

Existing literature has discussed the enablers of project integration from different dimensions. Halfawy and Froese [67] believed that the integration of multidisciplinary project processes throughout the project lifecycle can be achieved based on an integrated project system. Rutten et al. [75] indicated that the systems integrator undertakes the responsibilities of designing and producing CoPS (complex product systems) and adds value through system integration, thus playing a role in establishing and coordinating inter-organizational innovation in construction. Braglia and Frosolini [76] stated that the project management information system can be integrally implemented in extended enterprises to manage complex projects by adopting shared communication, common standards, and appropriate software tools for managing supply chains. The results of Zhang et al. [77] revealed that leadership styles have a mediated effect on the relationship between emotional intelligence and collaboration satisfaction in an integrated team. Oppong et al. [78] pointed out that a collaborative integrated project solution can be achieved through integrating the diverse needs, interests, and objectives of stakeholders into the design of a project. The empirical results of Shen et al. [79] verified that formal practices and social norms can improve interface management behaviors and achieve communication and coordination between different parties in EPC projects. However, most of the research on project integration

focuses on the integration of a certain dimension, while the research on promoting systemic project integration in the digitization of construction projects is still lacking.

### 2.2.2. Project Governance

Although the existing literature does not explicitly describe the relationship between project governance and project integration, the close relationship between them is indirectly reflected in literature [80]. Project governance can form cooperation and consistency among participants through contractual and relational governance mechanisms [81,82]. Contractual governance controls and coordinates the expected behavior of participants through formal rules, terms, and procedures. It sets out principles, general procedures, and primary responsibilities for all participants to guide the accomplishment of tasks; it integrates resources and maintains collaboration to achieve valuable creations [83,84]. Relational governance is an informal mechanism that enhances the social ties of participants by forming relational norms and trusts [81,85]. By sharing norms and values among participants and cultivating mutual trust, it can promote the coherence of partner interests and reduce opportunistic behaviors [86]. It is, therefore, beneficial to the implementation of planning and the achievement of consistency in the project process [82]. Thus, project governance can establish coordinated actions of different parties in implementing digitization, through formal or informal means, to facilitate project integration.

A project governance model provides comprehensive and consistent methods to control the project based on contractual and relational governance mechanisms. Considering a project as a nexus of both internal and external treaties that is governed by a structure of organizational arrangements, Winch [87] described the project governance model as a three-level system that includes the institutional level, the governance level, and the behavioral level. The institutional level set the 'rules of the game' in the project environment, thereby reducing uncertainty in organizational and individual decision-making. The behavioral level includes how managers typically respond to tasks. The governance level mediates between the institutional level and the behavioral level, and it includes the tectonic approach and process of an organization.

Based on the literature review and the governance framework of Winch [87], this study established a project governance framework for integration in digitization that displays the relationship between project governance, project integration, and digital technology, as shown in Figure 1. Project governance consists of the institutional level, the organizational level, and the behavioral level, which provides appropriate organizational conditions for the adoption and implementation of digital technology. Digital technology—commonly in the form of combining BIM with other technologies—provides technical support for effective inter-organizational governance. The interaction of the two facilitates the realization of BIM-based project integration to achieve efficient and integrated project management during the project lifecycle, as well as increase the value of co-creation among stakeholders.

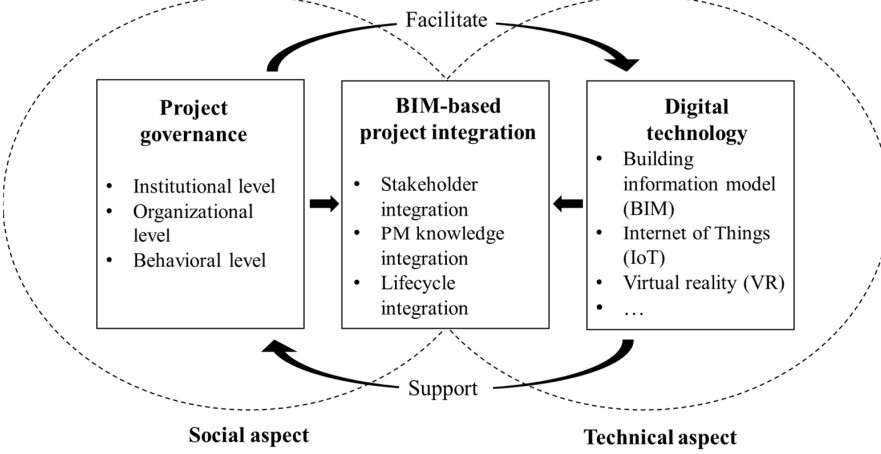

**Figure 1.** The project governance framework for integration in digitalization.

## 3. Methodology

### 3.1. Research Design

As the case study is the research strategy that allows in-depth analysis and understanding of complex social phenomena by incorporating a variety of evidence [88], this study adopted a single case study method to explore the project governance model for achieving integration in the digitalization of construction projects. We conducted the case study using participant companies in the case project as units of analysis. Based on the project governance framework, an in-depth analysis of this case project was conducted to obtain a deep understanding of project integration and the digitization of construction projects, as well as to explore ways of achieving project integration in digitization. Data were collected from extensive sources, including document review, site observation, and focus group interviews. The validity and reliability of data were verified through triangulation. Content analysis was then used to analyze data and develop the theoretical model since it is a data analysis method that provides an objective and systematic approach to make valid inferences from verbal, visual, or written data to describe specific phenomena [89]. Finally, the theoretical model developed from the case study was further cross-compared and discussed with the results of the literature review to achieve theoretical generalization.

### 3.2. Case Selection

In this study, the Inner Mongolia Minority Cultural and Sports Centre Project (abbreviated "CSC project" hereinafter) was selected as the case, which is in Hohhot, China. It was developed to celebrate the Inner Mongolia Autonomous Region's 70th-anniversary ceremony and, later, to hold international minority sports and large international horse racing events. The CSC project is a large and complex construction project. It has a total budget of 107 million Euros and covers an area of 80,000 m$^2$, including the main building, standard racetrack, and parking plaza. The main building consists of three parts: the grandstand building, the horse-showing building, and the multifunctional building. To present Mongolian ethnic characteristics, the design of the main building contains multiple shaped surfaces (e.g., round, oval, and hyperboloid) composed of aluminum plate and curtain walls, as well as deformed steel structures. Besides, the CSC project involved more than 20 main participants and a significant amount of coordination work between them.

Although the project is technically and organizationally complex, it was required to be completed in an extremely tight schedule (18 months) with industry-leading quality. Therefore, the project management company (the single general contractor in the project) attempted to adopt digital technology to complete the project. This means that it needs to deploy digital technology in the project under extreme constraints, taking full advantage of digitalization to fulfill project requirements. Thus, the project management company decided to strategically combine project integration management and digital technology organically to integrate stakeholders and their resources to support the realization of the project objectives. In the end, the project was not only completed within the time limit but also obtained significant cost-saving and high stakeholder satisfaction. In summary, this project has good applicability to the research questions. Inductive research of this case will also provide theoretical guidance for the digitalization of other construction projects.

### 3.3. Data Collection and Analysis

#### 3.3.1. Data Collection

The case data were collected from multiple sources, including document review, site observation, and focus group interviews. In terms of the document review, project documents, including project process reports, meeting minutes regarding milestones and major issues, commercial publications, and research papers were collected by the researchers. Basic information about the case project, including project organizational structure, internal and external stakeholders, project process, and project performance indicators was attained by reviewing those documents, which then provided a project context for the following site visit and interview survey.

With the consent and support of the PM team, the researchers conducted two rounds of site observation, from September 2017 to August 2019, to obtain first-hand data on the project. To avoid retrospective bias and self-censoring, two researchers carried out informal interviews with multiple informants (e.g., frontline managers and workers) separately to verify the document data and add qualitative and unfiltered impressions from the project site [88]. Besides, several demonstrations of the application of digital technologies were also organized at the project site.

Based on the result of the document review and site observation, a focus group interview survey was designed to collect in-depth data about the case project [90]. There were seven face-to-face semi-structured interviews that were conducted with a total of 25 interviewees, with each interview containing a theme related to the project's digitalization strategy and a corresponding interview outline (as shown in Table 1). The participants of each interview were selected according to the theme, and they came from various project stakeholder organizations and played an important role in the implementation of digitalization. The interviewees of each organization consist of at least a leader, manager, and executor to ensure the authenticity and comprehensiveness of the information received. Each interview lasted for 1 to 3 h and was recorded and interpreted into text with the interviewees' consent.

To ensure data validity and reliability, this study used a 'triangular verification' method [86], comparing data from site observation, document review, and interview survey; the discrepancies between the data were verified and explained through follow-up telephone interviews with the interviewees.

**Table 1.** Key information of focus group interviews.

| No. | Role and Number of Interviewees | Specific Interview Topic | Duration |
|---|---|---|---|
| 1 | Project client (2) | Project objectives and the motivation for adopting digitalization strategies | 1.5 h |
| 2 | Project management company CEO (1) | The organizational design of the project in implementing digitalization | 1 h |
| 3 | Project manager/ Deputy project manager (3) | The coordination of project processes and resources | 2 h |
| 4 | Project technology director (1) | The application of digital tools and platforms in the project | 1 h |
| 5 | Project management team members (5) | Management methods for implementing digital technologies in project tasks | 2.5 h |
| 6 | Design company, consulting company, contractor, subcontractors, and suppliers (10) | The participation and collaboration of stakeholders in the digitalization of the project | 3 h |
| 7 | Future users of the project (3) | Experience in using the project | 1 h |

### 3.3.2. Data Analysis

A content analysis approach was adopted in this study to analyze all the qualitative data collected. According to Bengtsson [91], the content analysis was performed through four steps—namely decontextualization (open coding), recontextualization, categorization, and compilation—so that qualitative data can be encoded and classified to develop the theoretical model. First, two researchers read all the material separately to identify elements of project governance and label them with a code under each of the three dimensions of integration. Second, two coding results were compared and checked, and differences were eliminated, through discussion, to form a unified list of governance elements. Third, the research teams organized the elements into a systematic structure according to their conceptual relationships and roles in the project governance framework. Finally, the analysis results and explanation were presented in this study. Data analysis and research method are summarized in Figure 2, which is adapted from Sting and Loch [92], in which they describe the complete process of data analysis in the case study.

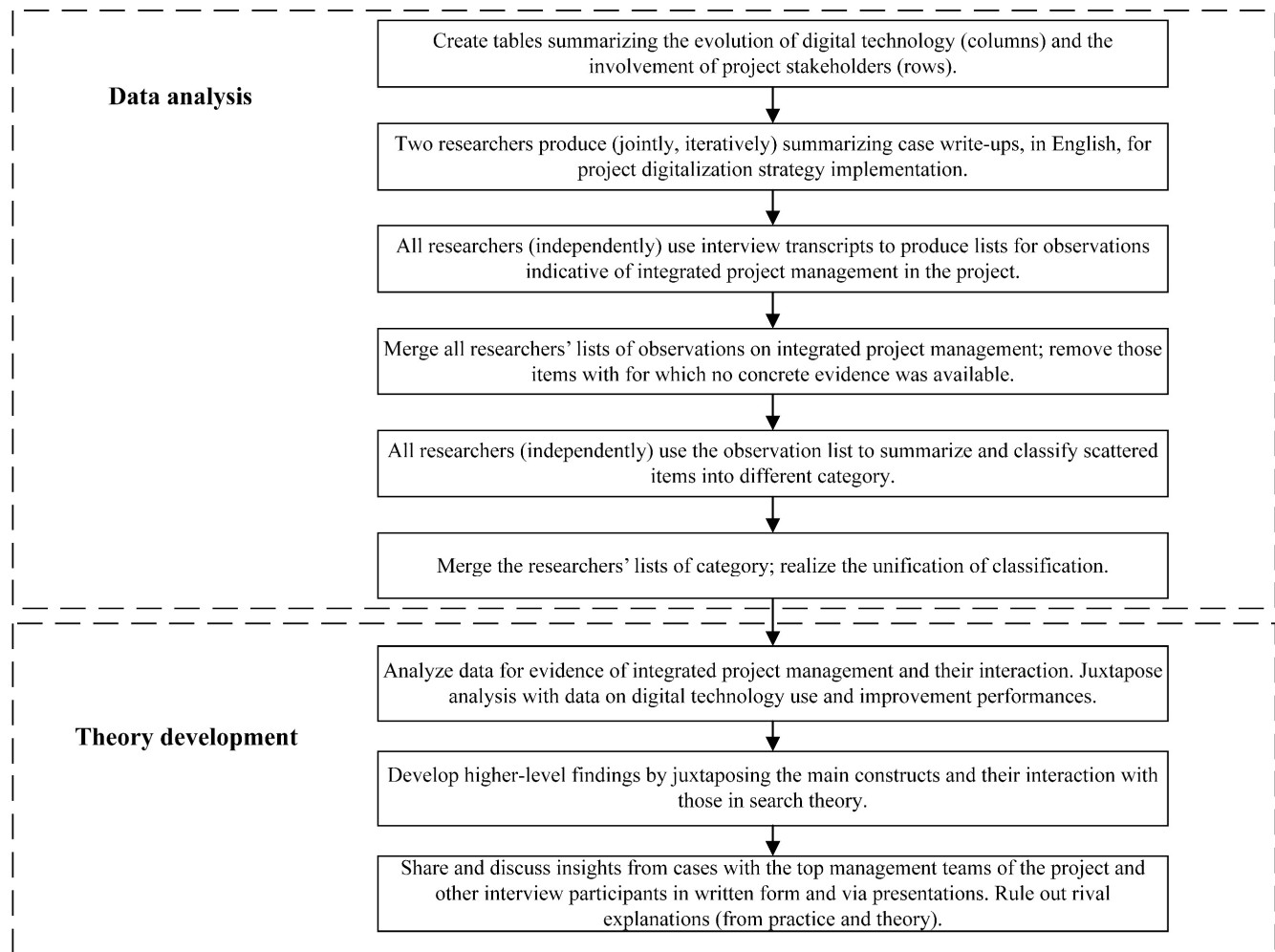

**Figure 2.** Detail process for data analysis and theory development.

## 4. Analysis and Results

### 4.1. Integration of Stakeholder

### 4.1.1. Clear Roles and Responsibilities

The CSC project adopted the EPC procurement model to select an experienced project management company as the single general contractor. The project management company (PMC) is responsible for the entire process of project design, procurement, construction, and commissioning, which provides the basis for the PMC's role as a system integrator. The PMC also clarified its roles and responsibilities at the beginning of the project to consolidate its system integrator role in the project stakeholders' network. Going beyond the basic goal of meeting contractual requirements in traditional project management, it set the realization of stakeholder needs and the creation of project value as its core goals; it assigned itself the threefold responsibility of a manager, consultant, and coordinator. As a manager, the PMC will perform the responsibility of overall project management within the scope of the client's authority; as a consultant, it will use its managerial experience and technical knowledge to solve problems and provide a variety of alternatives for the client and other stakeholders; as a coordinator, it utilizes the intermediary and coordinating role to punctually discover and solve the conflicts of stakeholders' goals and requirements in the digitalization process, thereby coordinating stakeholders to reach a unified digitalization goal.

As the PMC had a clear understanding of its role and undertook its responsibilities actively, it provided a prerequisite for the integration of stakeholders in the future. According to the project manager, "*By establishing our roles and responsibilities in implementing the digitalization from the beginning, we avoided role positioning bias, such as overstepping and under-performance, laying a solid foundation for integrating stakeholders and realizing value creation for them.*" To establish a coordinated stakeholder network to implement digitalization, the PMC then assigned the roles and responsibilities of main stakeholders in digitalization at the early stage of the project. Table 2 shows the responsibility matrix for BIM implementation.

**Table 2.** Responsibility matrix for BIM implementation.

| Project Stage | Stakeholders and Their Responsibilities | | | |
| --- | --- | --- | --- | --- |
| | Client | Design Firm | Project Management Firm | Main Contractor |
| Design | —— | Support | Main responsibility | —— |
| Construction | —— | Support | Management | Main responsibility |
| Completion | —— | —— | Management | Main responsibility |
| Operation and maintenance | Support | —— | Main responsibility | —— |
| Promotion and exhibition | Management | Support | Main responsibility | Support |

### 4.1.2. Digital Leadership

Despite the fact that PMC has made trials of integration management practices in a few projects, it has not combined integration management with BIM technology in large and complex project settings. Thus, this innovative digital scheme cannot be realized without strong leadership to drive this technological and organizational change. To ensure the sufficient capacity of the leadership to execute the digital strategy, top leaders were elaborately selected by the PMC internally and externally. The CEO of the PMC was appointed as the prime project leader, who is responsible for the allocation of digital resources to the project and the communication and coordination with the stakeholders; the deputy GM of the company, who is experienced in integration management practices, acted as the project manager to manage the process of digitalization; the PMC also introduced BIM technical experts from China Building Research Institute to serve as the deputy project manager and technical director, providing technical support for the BIM implementation. Equipped with other business executives, a project management team (PMT) was established with sufficient managerial and technical capabilities to carry out digitization.

The PMT was then committed to creating shared visions of digitalization among the project team. Top managers explained to project members the overall digital goals and strategies of the project, the ideas of integration management methods, and the application of BIM. Along with other training programs, a unification of project team members' mindsets was achieved to guide the integration of stakeholders in the digital process. As one of the PMT members said, "*after training, project team members reached a consensus of digital visions and strategies. They learned which processes can create value for stakeholders and the application of BIM in the context of integration management to improve efficiency. This knowledge has laid a good foundation for us to gain recognition from stakeholders and to motivate them to engage in digitalization*".

To promote stakeholders' acceptance of digitalization, project leaders elaborated, to the client and other stakeholders, the essential role of digitalization in realizing the project objectives and delivering value for them. After that, professional and technical personnel were assigned to train contractors and subcontractors to improve their BIM application capabilities. Besides, by exerting its technical capability, the PMT continually helped stakeholders make full use of BIM technology and promoted them to embrace digital practices. For instance, it regularly shared BIM experience and results with stakeholders, took the initiative to put forward more than 1000 design optimization suggestions for

designers based on the BIM model, and encouraged the construction contractor to utilize BIM for construction simulation, overcoming a few construction difficulties. As a result, the leadership of the PMT not only built up the confidence of all participants to collaborate in digitalization but also made them more willing to accept the management of the PMT.

### 4.1.3. Alignment of Stakeholders' Needs and Interests

To achieve the commitment of all stakeholders to digitalization, it is important to align the diverse needs and interests of different stakeholders with the objectives of digitalization. At the beginning of the project, the interests of the client were aligned with the PMC. The PMC believed that its economic benefits should come from the sharing of value creation and cost savings for the client, rather than adding additional management costs. The idea was welcomed by the client and, then, enshrined in contract terms such that the PMC can take 10% of the cost savings for the client as a share.

After obtaining the client's authorization, the PMT strategically aligned the stakeholders in the project. Since the needs and expectations of stakeholders will change dynamically with different stages of the project, the PMT collected, updated, and managed the needs of various stakeholders through comprehensive communication mechanisms. Then, the needs of stakeholders were linked to the project objectives and reflected in the BIM model by updating their corresponding tasks. However, due to the large number of stakeholders in the project, the requirements and interests of stakeholders often conflicted during the project lifecycle, which posed challenges to satisfying those requirements. An effective method adopted by the PMT was to distinguish and prioritize rigid requirements. According to the project technical director, "*it is important to reach understanding on key issues, which requires the engagement of all the stakeholders*". Based on this, the PMT resolved conflicts between the needs of stakeholders by leveraging its intermediary position and influence. Specifically, if one party's need is the rigid one that must be met, the project manager will persuade the other party with flexible needs to make appropriate concessions in exchange for compensation for other needs from the satisfied party. Alternatively, the PMT also helped the concessional party create additional value as another compensation method by utilizing BIM and an integrated management approach. For example, the project schedule required by the client is tight but must be accomplished, which puts huge pressure on the contractor's progress and cost management, so the PMT persuaded the client to compensate the contractor by way of reward and advance payment. The PMT also helped the contractor use BIM to optimize construction processes, improve construction efficiency, and reduce the time delay caused by rework and low efficiency.

Consequently, supported by BIM and fully utilizing its system integrator role, the PMT integrated the scattered needs and interests of stakeholders into a complete and comprehensive digital goal.

### 4.1.4. Unified Practices and Norms of Digitalization

Unified practices and norms are the basis for achieving the integration of stakeholders and the linkages of tasks. To this end, the PMT compiled the implementation standards of digitalization at the project planning stage, including the BIM implementation outline, BIM implementation standard, integration project management outline, and project management manual, etc. To convert the digital strategy into reality, the integration project management outline was prepared to describe the framework and content of the project integration management. Based on that, a project management manual was formulated by the PMT as a tool of communication to unify the understanding and ways of working on digitalization among stakeholders. In addition, the PMT also compiled the BIM implementation outline to describe the objectives, content, and control of the BIM implementation, as well as relevant organizational structure and responsibility. Aiming at achieving data sharing among stakeholders and integrated utilization of BIM, the BIM implementation standard was further designed to clarify BIM lifecycle usage processes, model standards, and data storage and exchange standards.

While the guidance of digitalization has been developed, forming harmonized actions of stakeholders to digitalization is an ongoing effort throughout the project lifecycle. "*Each stakeholder has its own routines and ways of handling issues*", the project manager said, "*it is unrealistic to expect them to quickly adapt and master the digital way of working proposed in the implementation standards. In particular, some large contractors have their own work standards, so they may be aggressive and difficult to communicate in formulating and accepting project standards, because the modification of the standards will cause them to incur additional costs.*" Therefore, the PMT also adhered to the dynamic control of stakeholders during the project lifecycle. They referred to the standards of most companies when they formulated the standards and norms, and they also established formal and informal communication mechanisms to obtain feedback from stakeholders, thereby navigating their behaviors according to the implementation standards of digitalization. Under the guidance of the above standards, an atmosphere of digitalization was gradually formed in the project to promote the usage of unified digital practices and norms to implement the project among stakeholders.

*4.2. Integration of Project Lifecycle*

4.2.1. Project Lifecycle Planning

Different from the decentralized management, from design and construction to the operation stage, the integration of the project lifecycle makes decisions and plans for the project lifecycle in the early stage of the project. In the project planning and design stage, the PMT made use of BIM and other digital technologies to support the decision-making and pre-planning of the project lifecycle with the cooperation of major stakeholders such as the client, designer, contractor, and operator.

During the preliminary design phase, the PMT worked with the designer to construct and optimize the 3D model of the main building. The building's surface is a hyperboloid composed of glass curtain walls and an aluminum plate roof, making its production and installation highly complex. Therefore, the PMT and the designer simulated and optimized the partition of the surface on the BIM model, which improved the building's appearance, the standardization of the production, and the installation of components. By specifying the external and internal structure of the building, a complete virtual building scene was formed in BIM.

In the detailed design phase, the PMT combined other actors to examine and optimize the design. The 3D model of the building was demonstrated to the client so that the client can fully understand the appearance and space of the building and pinpoint their needs, thus effectively avoiding engineering changes. Besides, through auditing the building design with the contractor, the PMT found that the base design of the main building adopted an equivalent height without the consideration of the landform elevation difference, which increased the amount of earthwork by nearly 400,000 m$^3$. To solve this problem, the exploratory unit collected and relayed the landform data to the BIM to adjust the building base height. Through simulation and optimization, the amount of earthwork was minimized, and the balance between filling and excavation was achieved, saving about ¥30 million and avoiding an adverse impact on the environment. Furthermore, the operator from the operation stage also participated in the optimization of the building design, so the planning and requirements of the operation stage were also adequately considered in the design.

The construction plan was also determined in the design stage by visualizing and optimizing construction processes, based on digital technologies, to improve work efficiency. For example, to complete the project on schedule, the three parts of the main building were planned and constructed simultaneously. BIM and VR were applied to organize the construction site and resource scheduling to ensure orderly parallel construction. Besides, considering that the 12 month duration of the project includes 5 months of extremely low winter temperatures, the main parties jointly planned the construction sequence which adopted an "outside-in" construction sequence instead of a traditional "bottom-up" sequence. The advanced planning of construction sequencing enables outdoor construction

to be completed before winter, so workers can carry out indoor engineering with indoor heating during winter, thus promoting on-time project delivery.

Although a great deal of time was spent introducing digital technology at the early stage of the project, the pre-decision-making and lifecycle planning, which was based on BIM, effectively optimized and coordinated construction processes, leading to the significant improvement of participants' efficiency and the high-quality delivery of the project.

### 4.2.2. Lifecycle-Based Benefit Distribution

The application of digital technology increases the workload in the early stage of the project as well as the collaboration between stakeholders, so the traditional project benefit allocation mechanism, based on separate stages of the project, cannot effectively compensate the stakeholders who invest extra time and effort in digitalization. On the one hand, the designer, as the main actor in the design stage, invested a large amount of effort in digitalizing the project, while the relevant parties in the following stages enjoyed the facilitated conditions brought by digitalization. On the other hand, the digitalization and optimization of the design could not be achieved without the engagement of other parties from the later project phases.

"*No one wants to do hard work for no benefit*", said the deputy project manager, "*you need to keep track of everyone's contribution and distribute the benefit fairly*". To ensure that the stakeholders can continue to participate in the process of digitalization, the benefit distribution mechanism of the CSC project was transformed into a lifecycle perspective to fit with the application of digital technology through contract terms. With the support of the information system connected with BIM, the PMT distributed the benefit by recording the effort participants make in each task during the project lifecycle. The record was also used by the PMT to communicate with participants in monthly project meetings so that every party can gain an understanding of others' work and contributions to project tasks. This promotes an agreement on the importance of everyone's work among stakeholders and builds mutual trust, which leads to the compliance of each stakeholder to the benefit distribution mechanism and its continuous engagement with digitalization. Meanwhile, all stakeholders are allowed to use the digital resource of the project regardless of whether they have digital capabilities, which ensures the integrity of project data, and this extra expenditure will be compensated by the benefits generated by digitization.

One of the PMT members pointed out, "people tend to overemphasize the importance of their work, whereas in a project everyone's work is important. The new way of distributing benefits surprised the project participants in the beginning, but the digital platform helped us accurately document the effort made by each party, which is the basis for convincing them and building trust".

### 4.3. Integration of PM Knowledge

Integrated Information System

Due to the limitation of project resources, there are often trade-offs between PM knowledge areas. Without the integration of different PM knowledge, project management may only achieve local optimization within a certain area, rather than the best result for the whole project. To make digital technology coordinate and optimize different PM areas, the PMT integrated the information systems of time management, cost management, quality management, procurement management, risk management, HSE management, and communication management based on the BIM, thus establishing a unified digital platform. The digital platform realized the information linkage of various PM knowledge fields, thus supporting the decision analysis and overall optimization of the project.

To achieve project objectives, the PMT realized the integrated management of various knowledge areas based on the digital platform. After completing the preliminary design of the project, the client and PMT used BIM to accurately calculate the quantity of work and determine the scope and pricing of biddings; all the contracts were standardized and managed through the contract management system. In terms of procurement management, the information on materials and components derived from the BIM model was applied to support project procurement: the product database of suppliers was connected with the BIM model to perform the digital processing of materials, which improves the work efficiency and processing accuracy of suppliers; through the combination of BIM and IoT, the real-time monitoring of components was realized to ensure the quality and timeliness of supply.

In addition, the PMT worked with the contractor to analyze and simulate the construction solution and construction sequence based on the BIM model. By evaluating and comparing the performance of different solutions in terms of time, quality, cost, safety, etc., the optimal construction solution was selected and then used to generate the resource allocation plan, construction schedule plan, safety control measures, and risk management strategy of the project accordingly. In the management of the construction site, the construction site was optimized based on the BIM platform to facilitate construction and achieve sustainability. For example, the assembly site and transport channels were organized to reduce transportation costs and time. The needful temporary facilities were also rationally replaced by early completed permanent buildings and reusable materials to reduce unnecessary waste and damage to the local environment. For instance, the stable, completed early, functioned as the office of the PM department, and sheet plates were used to pave roads at the construction site. Furthermore, based on the integration of information, including site layout, construction sequence, and mechanical scheduling, VR technology was introduced to form a visual construction solution, which supports the client and contractor to intuitively understand the construction solution and effectively control the construction quality.

Through the analysis of case data, we summarized governance elements under each integration dimension and value-added project outcomes achieved by integration, which are displayed in Table 3. The results demonstrate that project governance tailored for digitalization facilitated a BIM-based integration, thereby achieving value creation for different stakeholders. According to the project governance framework, all identified governance elements in Table 3 were further classified into three governance levels, and their relationships were revealed. Consequently, a holistic project governance model for digitalization was established, as shown in Figure 3. In the model, the column shows the governance elements required for each dimension of integration; the row manifests the governance elements contained in each governance level in digitalization; the connection of each element reflects the influential relationships among the governance elements. The elements on the institutional level forge inter-organizational collaboration among stakeholders, who have an impact on the organizational governance elements and shape the project organization over time. The elements on the organizational level show the establishment process of an integrated project organization network under the guidance of the institutional level. They influence the choice of behavioral mode and operation method. The elements on the behavioral level indicate the specific operation of digitalization in project delivery to realize value-enhanced outcomes. Therefore, the model illustrates the mechanism of project governance to realize system integration and advance the implementation of digitalization in construction projects.

**Table 3.** Summary of case study results.

| Integration of Stakeholders | Integration of Project Lifecycle | Integration of PM Knowledge | Outcomes |
|---|---|---|---|
| *Clear roles and responsibilities* <ul><li>The PMC clarified its roles as a system integrator and the triple responsibilities of a manager, consultant, and coordinator.</li></ul> *Digital leadership* <ul><li>Top leaders created shared visions of digitalization within the project team and among stakeholders through training and multi-channel communications.</li><li>The PMT leveraged its technical capabilities to help stakeholders address technical issues in digitalization.</li></ul> *Alignment of stakeholder's needs and interests* <ul><li>The PMC aligned with the client through shared visions and a benefit-sharing contract.</li><li>The PMT collected and updated stakeholders' requirements to link them with the project objectives and reflect them in the BIM model.</li><li>The PMT coordinated the conflict between requirements by prioritizing rigid demand and compensating concessional parties.</li></ul> *Unified practices and norms of digitalization* <ul><li>The PMT compiled the implementation standards of digitalization at the project planning stage.</li><li>The PMT performed dynamic control of stakeholders based on communication mechanisms to navigate and harmonize the behaviors of stakeholders to digitalization.</li></ul> | *Project lifecycle planning* <ul><li>In the preliminary design phase, the PMT worked with the designer to construct and optimize the 3D model of the main building.</li><li>In the detailed design phase, the PMT combined the designer, client, contractor, and operator to examine and improve the design based on the BIM model.</li><li>The PMT worked with the contractor and suppliers to simulate and optimize construction solutions using BIM, VR, and laser scanning.</li></ul> *Lifecycle-based benefit distribution* <ul><li>The benefit distribution mechanism adopted a lifecycle perspective in which the PMT recorded the effort participants make in each task during the project lifecycle and distributed benefits based on this.</li><li>The record was used by the PMT to communicate with participants in monthly project meetings so that every party could gain an understanding of others' works.</li></ul> | *Integrated information systems* <ul><li>The information systems of various PM knowledge areas were linked to BIM, including schedule, cost, quality, procurement, risk, HSE, and communication.</li><li>In procurement management, BIM and IoT were used to generate and monitor the information about materials and components to support the management of procurement, production, and logistics.</li><li>The optimal construction solution of the project was selected by simulating and evaluating comprehensive performance in time, quality, cost, and sustainability.</li><li>The construction solution was used to generate the resource allocation plan, construction schedule plan, safety control measures, and risk management strategy of the project.</li></ul> | *Government* <ul><li>Public services were improved by providing leisure and sports venues for the public.</li><li>Promoting the sustainable development of the local economy by flourishing of the tourism and cultural sector.</li></ul> *Client* <ul><li>The project was completed on time and with less cost and higher quality.</li><li>Operating the venues more efficiently with the help of the digital platform.</li></ul> *PMC* <ul><li>Obtaining profits from sharing cost-savings with the client.</li><li>Gaining knowledge and experience in the utilization of digitalization to deliver projects.</li><li>The company received endorsements from other stakeholders which increased its reputation and the potential for getting new projects.</li></ul> *General contractor* <ul><li>Forming a complete digital project construction work scheme.</li><li>Productivity was significantly improved by using BIM to simulate and optimize construction processes.</li><li>Cultivating a group of long-term partners on digitalization for future projects.</li></ul> *Design company* <ul><li>Creating a digitalization design solution for the hyperboloid design of the steel structure.</li><li>Gaining extra benefits from the digital design solution.</li><li>Cultivating a group of long-term partners on digitalization for future projects.</li></ul> *End users* <ul><li>Experiencing advanced venues and facilities</li><li>Getting more job opportunities</li><li>The project site environment is effectively protected</li></ul> |

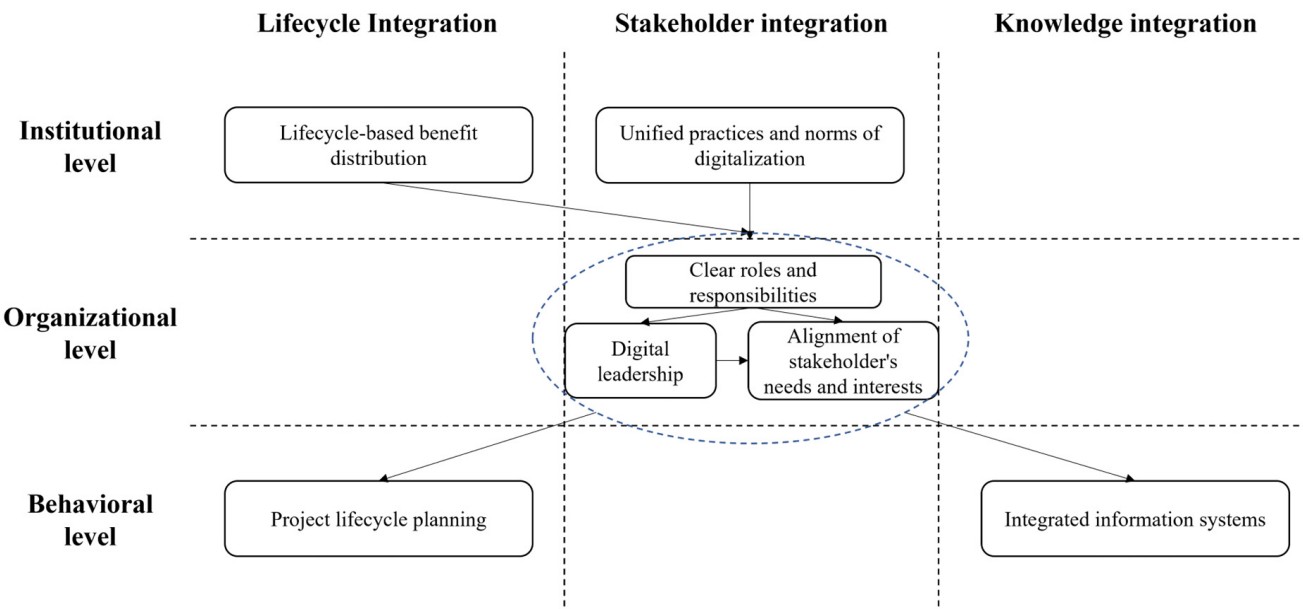

**Figure 3.** Project governance model for integration in digitalization of construction projects.

## 5. Discussion

The ultimate goal of digitalization in construction projects is better project delivery, which indicates enhancing project outcomes and creating value for stakeholders [8]. The digitalization of construction projects is a social-technical innovation system in which the adoption of digital technology will cause changes in activity links, resource ties, and actor bonds of the system, so the effectiveness of digitalization depends on how technical and organizational solutions fit into each other [73,93]. Without organizational adaptability, the adoption of digital technology may incur significant costs and fail to deliver the expected or even greater benefits for the project. The case study reported in this paper indicated that the organic combination of project governance and digital technology is a practical solution to realize project integration, thus improving outcomes of digitalization in construction projects. On the one hand, digital technology makes it possible to realize integration in construction projects. It can digitize the physical and managerial information in the project lifecycle and store, and it can present project data on an integrated digital platform, thus providing a feasible solution for project integration. On the other hand, project governance creates an integrated organizational environment for applying digital technologies throughout the project. It increases the acceptance attitude of digital technologies among stakeholders' network and supports the formation of coordinated and unified ways of cooperation. This drives the diffusion of digital technologies among the stakeholders, further supporting the realization of project integration. Therefore, effective project governance towards integration is a feasible organizational scheme to transform digitalization from a theoretical utopia into reality.

The results also show that a holistic BIM-based project integration was achieved through the interaction of three integration dimensions: stakeholder integration, lifecycle integration, and PM knowledge integration. First, stakeholder integration provided a foundation for lifecycle integration and PM knowledge integration as it established stakeholders' unified digitization goals and coordinated actions toward digitalization. This result is also reflected in literature that stressed the role of stakeholder management and collaboration in the digitalization of construction projects [20,32]. Second, the interplay of lifecycle integration and PM knowledge integration realized value creation for stakeholders and gave full play to the effects of digitalization. In the case study, each stage of the project lifecycle involved the integration of different PM knowledge areas, and all PM knowledge areas were planned, monitored, and controlled from a project lifecycle perspective, thereby

supporting optimal decision-making. The effects of lifecycle integration and PM knowledge integration on project value creation were also explained by scholars, respectively. Lifecycle integration connects discrete project processes to enable lifecycle decision-making at the early stage; it can also input the needs, knowledge, and expertise of project participants into the project design early to pursue value-enhancing project outcomes [61]. PM knowledge integration was also shown to improve PM practices by supporting the framing of integrated project plans and making globally optimal decisions on PM [56]. However, few studies have explored the interrelationships between the three dimensions of integration in promoting the digitalization of construction projects, which were revealed and explored by this research.

Furthermore, a three-level governance model was developed based on the literature review and case study to implement BIM-based project integration, which includes the institutional level, organizational level, and behavioral level. In terms of the institutional level, the benefit distribution mechanism was found to be an important prerequisite to realize lifecycle integration through both contractual and relational governance mechanisms. According to existing research, the imbalance between costs and benefits is a critical reason for the low enthusiasm of stakeholders in digitalization [94]. A fair benefit distribution mechanism is, therefore, required to align the benefits that stakeholders gain with the value they add to the project by quantifying and comparing the input and output factors [72,95]. The results of the case study indicate that the pricing method of contracts should be adjusted to develop a lifecycle-based benefit distribution mechanism in order to improve stakeholders' motivation. Stakeholders' contributions need to be tracked efficiently during the project lifecycle with the support of a BIM-based digital platform, which can be transparently shared by stakeholders to develop mutual understanding and trust. Therefore, the lifecycle-based benefit distribution mechanism can effectively ensure the fairness of benefit distribution among stakeholders and reduce conflicts, as well as encourage stakeholders to continuously participate in digitalization and undertake various tasks in the digitalization process.

Unified practices and norms of digitalization is another important element at the institutional level. Some contractors with digital capabilities had resistance when adopting unified standards at the initial stage, which is consistent with the conclusion that the conflict of standards is a hindrance factor pointed out in the existing research [96]. In this case, there existed competition among organizations, which would reduce the cost of adapting to the standard by striving for standard-setting power [97]. This study found that main stakeholders strive for the power to formulate digital technical standards based on their digital technology capabilities and leadership. Unified practices and norms of digitalization constrain the actions of all stakeholders, reduce the conflict in the project lifecycle, and ensure the effective implementation of digitalization.

From the perspective of the organizational level of project governance, contrary to the assumption of pursuing different strategies in the project [98], the case study showed how different organizations were coordinated through the project organization. The study found that effective leadership and coordination of project stakeholders are required when dealing with the complexity of the digitalization of construction projects. In the project organization, a PMT is usually established to supervise the project and guide the project implementation. This is considered the best practice of project management methods [99]. However, research shows that, when a project team composed of multiple stakeholders is responsible for guiding the project, the instructions given to the project may be conflicting and vague, which, in turn, will slow down decision-making and lead to project delay [100]. This case study provided a coordination mechanism for multi-stakeholders in the process of project digitalization. The implementation of integrating the stakeholders was accomplished by a system integrator [101]. It is indicated by researchers that system integrators can set up inter-organizational cooperation in construction projects [75,102]. By clarifying its roles and responsibilities at the beginning of the project, the system integrator can effectively avoid the ambiguity of responsibilities, providing a foundation to integrate stakeholders and execute digital strategies.

Furthermore, it is necessary to develop and utilize digital leadership to promote stakeholders' participation in digitalization. Digital leaders, such as the PMT of the case, played a vital role in inspiring organizational confidence in innovative but risky initiatives [103]. They were required to present a forward-thinking and proactive style to create a digital vision, and they had sufficient managerial and technical competence to motivate stakeholders to follow the vision [18,103]. Alignment of stakeholders' needs and interests is also required to promote the involvement of stakeholders in digitalization. The motivations of each project participant are often hidden or independent, so it is important to identify and align their needs and interests to facilitate inter-organizational collaboration [72]. By achieving alignment between digital goals and stakeholders' goals, stakeholders were motivated to achieve collaboration and actively engaged in digital strategies.

From the perspective of the behavioral level of project governance, through a unified institutional system and a coordinated governance organization, all stakeholders in the project began to gradually adopt unified behaviors and standards. The formulation of standards and the application characteristics of digital technology lengthened the time of the early planning stage of the project, while the efficiency improvement brought about by the application of digital technology greatly shortened the time of the project implementation stage. Therefore, the project planning and implementation stages are not isolated. The project needs not only a plan for the implementation stage but also a complete lifecycle plan. By involving project stakeholders early in project processes, the input of knowledge and expertise in the early stage was improved. It benefited the incorporation of various stakeholders' needs into the project design, as well as value-enhancing project outcomes. Furthermore, the decision-making and implementation of the project plan depend on the acquisition of project data, and the integrity and comprehensiveness of data acquisition affect the quality of project decision-making. A comprehensive information system and platform could aggregate various data in the project, thereby providing sufficient data volume for the application of project digital technology. Decisions made after synthesizing data from different PM knowledge fields can take into account the interests of various stakeholders, thereby making project work arrangements more scientific and rational.

## 6. Conclusions

The article explored how to promote the digitization of construction projects by project integration from a governance perspective. In recent years, only a handful of projects have indeed achieved digitalization and benefited from it because the decentralized project organization and mismatched project governance approaches hinder the usage of digital technologies. However, few researchers have studied project governance approaches, in the context of the digitalization of construction projects, from an organizational perspective. By conducting a literature review and a case study, this study concluded that project governance creates an integrated organizational environment for the implementation of digital technology, and digital technology provides the technical infrastructure for project governance; the interplay of project governance and digital technology can realize the comprehensive integration in construction projects, as well as the value-enhanced digitalization outcomes.

A conceptual project governance model was further proposed to manifest the mechanism of project governance to achieve three dimensions of integration in the digitalization of construction projects, namely stakeholder integration, lifecycle integration, and PM knowledge integration. In the model, governance elements are stratified based on the three governance levels—institutional level, organizational level, and behavioral level—and their relationships are also displayed. The institutional level consists of two elements, including lifecycle-based benefit distribution and the unified norms and practices of digitalization, which define the rules of project digitization to guide PM and forge inter-organizational collaboration among stakeholders to promote the adoption and implementation of digital technology in the stakeholder network. The organizational level is achieved through clear roles and responsibilities, digital leadership, and alignment of stakeholders' needs and

interests. It shows the establishment process of an integrated project organization network under the guidance of the institutional level. The behavioral level consists of project lifecycle planning and integrated information systems, which indicate the specific operation of digitalization in project delivery to realize value-enhanced outcomes.

The conceptual model proposed in this study has three main novelties compared with the existing project governance models. First, it is tailored for the digitalization of construction projects. Digitalization changes project organizations' ways of working, while existing project governance models are not applicable in the context of digitalization. The model in this study, however, illustrates a novel project governance solution for integration based on digital technology. Second, many existing project governance or management models focus on the institutional level or operational (behavioral) level, while this model looks at project governance from a holistic view and comprehensively identifies governance elements for digitalization from three governance levels. Third, the proposed model is systematic, as it reveals governance elements required for each integration dimension, as well as the influential relationships between those elements, which are not clearly displayed in previous models [96].

This study has both theoretical contributions and practical implications. Theoretically, this research fills the gap in the literature relating to how to promote the digitalization of construction projects from an organizational perspective. Taking project governance as a theoretical lens, this study established a conceptual project governance model for integration in the digitalization of construction projects. The model not only provides a theoretical foundation for governing construction projects to transform project organization networks to fit with digital technology, but also enriches the project governance theory and extends its application to the digitalization context.

In practice, the findings of this study contribute to the understanding and management of the digitalization of construction projects. The conceptual project governance model provides project managers with a practical approach to strategically implement digitalization in construction projects and maximize the use of digital technology to integrate various pieces of project information to achieve value-enhanced project outcomes. By facilitating the digitalization of construction projects, this study can also be conducive to the digital transformation in the construction industry.

This study is based on a single large complex construction project, so future research is recommended to test the proposed conceptual project governance model by using different large construction projects and exploring integrated governance mechanisms under digitalization in the other project types.

**Author Contributions:** Conceptualization, Z.L., R.D., Z.G. and O.E.; Methodology, Z.L., R.D. and Z.G.; Software, Z.G.; Formal analysis, R.D. and Z.G.; Investigation, Z.L.; Writing—original draft, Z.L. and Z.G.; Writing—review & editing, Z.L., Z.G. and O.E.; Supervision, R.D. and O.E.; Project administration, R.D.; Funding acquisition, R.D. and O.E. All authors have read and agreed to the published version of the manuscript.

**Funding:** This research was funded by National Natural Science Foundation of China under Grant number 72171134, and National Natural Science Foundation of Shandong Province under Grant number ZR2021MG037.

**Data Availability Statement:** Data is available on request due to restrictions such as data protection or ethical reasons. The data presented in this study are available on request from the first author. Data are not publicly available due to commercial use.

**Acknowledgments:** This research was funded by National Natural Science Foundation of China, and National Natural Science Foundation of Shandong Province.

**Conflicts of Interest:** The authors declare no conflict of interest.

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
