# Peer review of "Fostering Digitalization of Construction Projects through Integration: A Conceptual Project Governance Model"

_buildings, doi:10.3390/buildings13030825_

Round 1

Reviewer 1 Report

The authors chose "Fostering Digitalization of Construction Projects through Integration: A Project Governance Model" as the title of their study.

In the abstract, the authors claim that "The developed model will provide an approach for construction enterprises to promote project digitalization".

The authors also propose the following research objective," this paper aims to propose a project governance model for digitalization".

The problem is that the article does not include any models.

In addition, the authors do not use any mathematical modeling tool.

In my opinion, the article is an overview of the benefits of using BIM and integrated project delivery on the example of one project in China.

I did not find any scientific novelty or practical benefit in the article.

Author Response

Response to Reviewer 1

Comment - The authors chose "Fostering Digitalization of Construction Projects through Integration: A Project Governance Model" as the title of their study. In the abstract, the authors claim that "The developed model will provide an approach for construction enterprises to promote project digitalization". The authors also propose the following research objective," this paper aims to propose a project governance model for digitalization". The problem is that the article does not include any models. In addition, the authors do not use any mathematical modeling tool. In my opinion, the article is an overview of the benefits of using BIM and integrated project delivery on the example of one project in China. I did not find any scientific novelty or practical benefit in the article.

Reply - Thanks for your deep insights into our study. We agree with you that our research does not use any mathematical modelling tools. But a conceptual model for project governance was developed in this research based on qualitative research methods, including a literature review and case study. The data collection process in qualitative research requires information on case phenomena, and the data are collected from multiple sources of evidence and analyzed non-statistically. Accordingly, we first collected the fundamental elements of project integration and project governance through a literature review. Then a typical project with a large regional impact was used as a case to identify and verify relationships between these elements, through a rigorous case analysis process (the research process has been described in Section 3.3). Finally, a conceptual model of integrated governance model for the digitalization of construction projects was constructed which can help to guide projects with digitalization needs.

We also agree with you that this article presents the benefits of BIM and integrated project delivery, while this is only one of the contributions of our research. The implementation of digitalization is highly complex in construction projects, so it is difficult to achieve it by applying digital technology or delivery method separately. In our previous interviews with project managers and employees on different construction projects in countries like China and the UK, many indicated that they rarely adopted digital technologies such as BIM strategically in their projects and that they lacked sufficient incentives and benefits to drive them to do so. Thus, our research aims to address this issue by focusing on how to achieve an effective combination of digital technology and project management to release the full benefits of digitalization through integration. This point is rarely mentioned and addressed in existing research and is the major novelty and practical benefit of this research. This research explored the project governance approach to realizing mutual promotion and support of project management and digital technology, that is: integrated project management and delivery promote the implementation of digital technology in projects, and the implementation of digital technology (BIM) improves the efficiency of project management. The theoretical value of this research is that the proposed digitalization model reveals the relationship between various management integrations in existing research from the perspective of project governance, and how they worked in project digitalization. From a practical point of view, this research can be used by construction companies to govern construction projects in the process of project digitalization and to realize value co-creation and better project delivery.

Thank you again for your insightful comments and we hope that the above explanation can fully respond to your comments.

Reviewer 2 Report

Dear Authors , 

Good day to you. It's Great Work and an essential contribution toward the Digitalization of Construction Projects. Nicely and well structure formulated. 

Title : Fostering Digitalization of Construction Projects through Integration: A Project Governance Model

Abstract : The construction industry has fared poorly in the process of digital transformation, while the main challenge is the digitalization of construction projects. Changes in project management approaches are urgently required in construction organizations to better align digital technology and organizational conditions. However, little literature has explored the pivotal role of the project management approach from an organizational perspective. To fill this gap, this research investigates the ways of using an integrated governance model to promote the digitalization of construction projects through a case study. Three integration dimensions, namely stakeholder integration, lifecycle integration, and PM knowledge integration are identified, and their attributes are displayed and further stratified based on the three levels of the governance model, including institutional level, organizational level, and behavioral level. The logical relationship between attributes and their roles in project digitization is finally summarized. The developed model will provide an approach for construction enterprises to promote project digitalization.

A few observation has been made as follows : 

1. Methodology part Needs to be cross check that is not described well enough. 

2. Literature review has done in a proper way. 

3. Obejcetive of the paper is not well displayed in paper that needs to be address. 

4. Conclusion is written in general way needs slight infrenec sof results what we got in research. 

5. Reference formatting is not maintained uniformity throughout and last references is not cited in the body of the paper. 

6. Cross check all the citataions 

Best wishes. 

Author Response

Response to Reviewer 2

Comment - Good day to you. It's Great Work and an essential contribution toward the Digitalization of Construction Projects. Nicely and well structure formulated. 

Reply - We are thankful to Reviewer 2 for providing us with valuable feedback on our paper. We also thank you for the encouraging comments on the outline and contribution of the paper. We believe that the content and the value of the paper have been further improved by incorporating your feedback. Please see our detailed answer below.

Comment 1- Methodology part needs to be cross check that is not described well enough. 

Reply 1- Thanks for your valuable comment. We have made significant changes to section 3. Methodology in terms of content and structure.

Firstly, in section 3.1 Research design, in addition to the introduction of the research methods, we added an introduction to the whole process of conducting a case study in this study, including the process and method of data collection, data analysis and theory development. It helps readers to better understand the role of each part in section 3. “Based on the project governance framework, an in-depth analysis of this case project was conducted to obtain a deep understanding of project integration and digitization of construction projects and to explore ways of achieving project integration in digitization. Data were collected from extensive sources, including document review, site observation, and focus group interview. The validity and reliability of data was ensured through triangular verification. Content analysis was then used to analyze data and develop the theoretical model since it is a data analysis method that provides an objective and systematic mean to make valid inferences from verbal, visual, or written data so as to describe specific phenomenon [89]. Finally, the theoretical model developed from the case study was further cross compared and discussed with the results of the literature review to achieve theoretical generalization.

Secondly, in section 3.2 Case selection, based on the requirements of single case study, we supplemented the introduction the typical characteristics of the case and its matching with the objectives of this study, and explained the reasons for choosing this case as the research object. “Therefore, the project management company (the single general contractor in the project) attempted to adopt digital technology to complete the project. This means that it needs to deploy digital technology in the project under extreme constraints and takes full advantage of digitalization to fulfill project requirements. Thus, the project management company decided to strategically combine project integration management and digital technology organically to integrate stakeholders and their resources to support the realization of the project objectives. In the end, the project was not only completed within the time limit, but also obtained significant cost-saving and high stakeholder satisfaction. In summary, this project has good applicability to the research questions.

Thirdly, we adjusted the structure of 3.3. Data collection and analysis to integrate the data collection process into one subsection and the data analysis process as a separate subsection. In section 3.3.1. Data collection, we have refined the content of this section and introduced the work we carried out at this section more clearly. In section 3.3.2 Data analysis, we have added contents of data analysis methods and processes. This subsection describes the steps of we use content analysis method to building a theoretical model from obtaining data from the project. “A content analysis approach was adopted in this study to analyze all the qualitative data collected. According to Bengtsson [92], the content analysis was performed through four steps - namely decontextualization (open coding), recontextualization, categorization, and compilation - so that qualitative data can be encoded and classified to develop the theoretical model. First, two researchers read all the material separately to identify elements of project governance and label them with a code, under each of the three dimensions of integration. Second, two coding results were compared and checked, and differences were eliminated through discussion to form a unified list of governance elements. Third, the research teams organized the elements into a systematic structure according to their conceptual relationships and roles in the project governance framework. Finally, the analysis results and explanation were presented in this study. Data analysis and research method are summarized in Figure 2, which is adapted from Sting & Loch [90], in which, they describe the complete process of data analysis in the case study.”

Finally, we've made minor modification to some sentences and words to make our descriptions more accurate.

Comment 2- Literature review has done in a proper way. 

Reply 2 - We are very appreciated that our works on literature review have got your approval.

Comment 3- Objective of the paper is not well displayed in paper that needs to be address. 

Reply 3 - The authors have thoroughly revised it and the objective of the paper is stressed in the 4th paragraph of the 1. Introduction. Actually, the 1.Introduction part has already proposed the research objective, while we present the objective in a clearer way after the revision, as shown in Line 4, paragraph 4, 1.Introduction: “Therefore, the purpose of this study is to address the organizational barriers of digitalization and reduce the misalignment of project organizations and digital technology by adopting a project governance approach, so as to improving the application performance of digital technology in construction projects. To achieve this, the following research questions are posed: 1) What is the relationship between project governance, project integration, and digitization? 2) How to achieve project integration to provide a favorable environment for the application of digital technology through a project governance approach, thereby promoting the digitalization of construction projects?

Comment 4- Conclusion is written in general way needs slight inferences of results what we got in research. 

Reply 4 - Noted and worked with thanks. In the 1st paragraph of 6. Conclusion, we added the conclusions that can be obtained based on the research results. this model on how organizations and technologies are effectively promoted in project digitalization. “By drawing attention to the case project success and its experience on project digitalization, this study argued that project integration provides an environment for the implementation of digital technology, and digital technology creates conditions for project integration, and they are both indispensable for integration. Besides, the results of the case study show that successful project integration can be achieved from project governance which aligns the project stakeholder, lifecycle, and PM knowledge.” These conclusions respond to the research objectives we put forward in the 1. Introduction.

Comment 5- Reference formatting is not maintained uniformity throughout and last references is not cited in the body of the paper. 

Reply 5Thank you for spotting this oversight which we have now revised to maintain conformity throughout the paper. All references are now cited.

Comment 6 - Cross check all the citations 

Reply 6Noted and worked with thanks. During the revision process, every author checked all citations, and the problems with citations have now been effectively revised.

Reviewer 3 Report

1. The resolution of figure 1 is not high enough. Please redraw high-resolution figures 1-3.

2. More related works about big data, information system, and construction building may be cited, such as

[1] Chaurasia, Sushil S., and Surabhi Verma. "Strategic determinants of big data analytics in the AEC sector: a multi-perspective framework." Construction economics and building 20.4 (2020): 63-81.

[2] Huang, Qian, et al. "Smart building applications and information system hardware co-design." Big Data Analytics for Sensor-Network Collected Intelligence. Academic Press, 2017. 225-240.

3. In Section 4.3, how about adding one more sub-section to compare this work with existing models in the literature?

Author Response

Response to Reviewer 3

Comment 1- The resolution of figure 1 is not high enough. Please redraw high-resolution figures 1-3.

Reply 1 - Thank you for reviewing our paper so carefully. According to your suggestion, we redrew the three pictures, which now is high-resolution.

Comment 2 - More related works about big data, information system, and construction building may be cited, such as

[1] Chaurasia, Sushil S., and Surabhi Verma. "Strategic determinants of big data analytics in the AEC sector: a multi-perspective framework." Construction economics and building 20.4 (2020): 63-81.

[2] Huang, Qian, et al. "Smart building applications and information system hardware co-design." Big Data Analytics for Sensor-Network Collected Intelligence. Academic Press, 2017. 225-240.

Reply 2 - Noted and worked with thanks. The two papers you recommended have been cited to the appropriate places in the paper. Moreover, we have cross-checked the citations of the paper, added several additional relevant literatures, which makes our paper more scientific and rational.

Comment 3- In Section 4.3, how about adding one more sub-section to compare this work with existing models in the literature?

Reply 3 - Thanks for your deep insights into our study and pointing out the problems regarding the organization of section 4.3. In Section 4, we mainly introduce the process of case analysis and model construction, of which Section 4.3 introduces the elements and structure of the model. Therefore, we believe it would be better to added it in 5. Discussion. At the 2nd paragraph of 5. Discussion, we introduced the changes of our model compared with other models, which are mainly reflected in the new features reflected by the meaning and logical relationship of elements. “Moreover, the meaning and logical relationship of their elements have changed in the context of digitalization, as shown in the table 3. These changes are mainly manifested in three characteristics: first, digital technology - oriented, such as the way of benefit distribution, the formation of leadership; Second, holistical, which is manifested in the correlation between the elements based on digital technology; Third, hierarchical, reflected by the progressive logic of these elements in project organization construction and digital technology application. Therefore, from the perspective of project governance, all these elements are integrated into a governance model and further divided into three levels: institutional level, organizational level, and behavioral level.” At the last paragraph of 5. Discussion, we introduce the difference between our model and other models from the perspective of element relationship again, and also explain the relationship between this model and other project governance model. “In summary, the project governance model established in this study integrates the main elements of the existing integrated project management model, such as stakeholder integration management, lifecycle integration management, etc., More importantly, it establishes the connections between different types of elements, so as to realize the organic integration of the existing integration management model as a whole. Any change of the element will impact on whole model through their relationship. From the perspective of project governance, this study is an extension of Winch's project governance model in the research of project digitalization. In contrast, the model proposed in this study clarifies the main contents of different levels of project governance under the background of project digitalization, and further enriches the project governance theory.

Round 2

Reviewer 1 Report

I suggest the authors specify exactly what type of model they propose in their research. Add "conceptual model" to the article title and abstract.

I also think that it is necessary to change the type of article to "case study".

Author Response

Response to reviewer

Comments: I suggest the authors specify exactly what type of model they propose in their research. Add "conceptual model" to the article title and abstract.

I also think that it is necessary to change the type of article to "case study".

Reply: Thanks for your valuable comments. Per your request, we have revised the title and abstract, and added "conceptual" to the article title and abstract. Meanwhile, in the main body of the paper, we also modify and explain the type of model. For example, in the last paragraph of 5. Discussion. “In summary, the conceptual project governance model established in this study integrates the main elements of the existing integrated project management model”, and in the 2nd paragraph of 6. Conclusion “A theoretical model was further proposed to reveal the mechanism of the project governance model in the digitalization of construction projects.”

Besides, we have changed the type of article to "case study".
